# Effects of dry cupping on exercise, autonomic activity and sleep in baseball players during preseason and in-season conditioning

Chien-Liang Chen[1]*, Jing-Shia Tang[2]

1 Department of Physical Therapy, I-Shou University, Kaohsiung, Taiwan, 2 Department of Nursing, Chung-Hwa University of Medical Technology, Tainan, Taiwan

* chencl@isu.edu.tw

## Abstract

### Background

Cupping therapy has been shown to alleviate muscle fatigue, sustain exercise capacity, enhance post-exercise recovery of autonomic activity, and improves sleep quality. However, variations in athletes' training intensity, competition pressure, and fatigue levels throughout a sports season remain underexplored. Few studies have investigated whether the health benefits of cupping differ across various phases of a sports season. This study aimed to examine the effects of short-term cupping on athletes during preseason conditioning (PSC) and in-season conditioning (ISC).

### Methods

Forty university baseball players were recruited and randomly assigned to either the cupping (dry cupping at − 400 mmHg) or sham (dry cupping at − 100 mmHg) group. Cupping was applied to the upper back and shoulders for 15 minutes, twice a week for 8 consecutive weeks (4 weeks each during PSC and ISC).

### Results

Cupping had no significant effect on upper-extremity function during either PSC or ISC. Exercise tests during PSC and ISC revealed no postintervention changes in peak power, peak oxygen consumption, or anaerobic threshold. However, cupping during PSC improved postexercise recovery of low-frequency power (LF; $P = .013$; a component of heart rate variability) and that during ISC improved recovery of the LF/high-frequency power ratio ($P = .004$) and LF% ($P = .037$). Additionally, cupping during PSC notably enhanced daytime function, as measured by the Pittsburgh sleep quality index ($P = .026$).

**Data availability statement:** All relevant data are within the manuscript.

**Funding:** This research was financially supported by the Ministry of Science and Technology of the Republic of China under contract MOST 107-2314-B-214-002.

**Competing interests:** The authors have declared that no competing interests exist.

## Conclusions

The benefits of cupping therapy vary between PSC and ISC. Cupping during PSC and ISC notably improved the postexercise recovery of autonomic and sympathetic activities, respectively. However, improvements in sleep quality were only observed during PSC.

## Introduction

In various phases of a sports season, athletes experience different levels of stress and fatigue [1]. Thus, similar healthcare measures may lead to different outcomes [2]. Year-round conditioning can be divided into three key periods as follows: off-season conditioning, preseason conditioning (PSC), and in-season conditioning (ISC). PSC and ISC necessitate high training intensity and specialized care. The aim of PSC is to maximize utilization of the energy system specific to the sport, whereas that of ISC is to maintain their fitness level throughout the competition cycle [3]. Inadequate recovery during high-intensity training may result in adverse health effects owing to exercise fatigue, ultimately reducing their competition performance [4,5]. Problems commonly associated with overtraining include autonomic imbalance, sleep disturbance, poor performance, and persistent fatigue [6,7]. Ensuring the optimal efficiency of recovery from sports fatigue and autonomic nervous system (ANS) activity across the season is crucial for maintaining athletes' physical function and sports performance of athletes [8,9].

It is common for athletes to employ non-pharmacological interventions, such as alternating hot and cold-water therapy, cryotherapy, compression garments, electrical stimulation, and massage, to enhance recovery from training-induced fatigue [10]. However, these methods may not be as economical or convenient as cupping. Many athletes use dry cupping for pre-competition health care and in-season recovery. For example, the cupping marks on the back of the renowned swimmer Michael Phelps during the 2016 Rio Olympics garnered global attention. Studies have demonstrated the benefits of cupping for athletes [11,12]. Cupping mitigates muscle fatigue, maintains sports performance [13,14], promotes ANS activity recovery after intense exercise [13,15–17], enhances blood volume and tissue oxygenation [18,19], and improves sleep quality [20,21]. However, owing to differences in training intensity and competition schedule at different periods of the sports season, the extent of sports fatigue also varies significantly. Few studies have explored the changes in athletes' subjective perceptions and objective physiological responses due to cupping during conditioning at different periods of the season.

Although the benefits of cupping are known worldwide, no clear guidelines for cupping intervention currently exist. Many uncertainties remain regarding the operating principles of cupping for specific sports groups, such as intervention areas, duration, and time points. This study aimed to investigate the short-term benefits of regular cupping for university baseball players during different conditioning periods. We hypothesized that regular dry cupping applied to the upper body muscles (trapezius and deltoids), commonly used during pitching and batting, would lead to improvements in subjective perceptions, such as sleep quality and upper limb and shoulder function, as well as objective physiological responses related to ANS activity during both the PSC and ISC periods.

## Materials and methods

### Participants

This study recruited 50 volunteers from a university baseball team for a recruitment period from 01/10/2018 to 15/01/2019. Considering participants from the same team regularly

engaged in collective training helped minimize inherent between-team differences in training modes, intensity, cycles, and competition schedules, which could have affected the effectiveness of the experimental intervention. Players who were unable to participate in exercise tests because of musculoskeletal injury and those who already have cupping habits were excluded. Finally, 44 participants were included. During the research process, 4 participants dropped out because of low attendance, and finally only the data of 40 athletes (age = 19.5 ± 0.2 years, height = 176.7 ± 0.8 cm, and weight = 80.4 ± 2.0 kg) were analyzed (Fig 1). The adequacy of the sample size for this study was evaluated using G-Power software (version 3.1.9.4, Heinrich-Heine-Universität Düsseldorf, Germany). An a priori power analysis for repeated measures analysis of variance (RM-ANOVA) was conducted, with the following parameters: an effect size of .60 [22], a statistical power of .80, and an alpha level of .05. The analysis indicated a minimum required sample size of 24 participants. Thus, the final sample of 40 participants exceeded the minimum requirement, supporting the statistical validity of the study. This study was approved by the Human Research Ethics Review Committee (approval number: NCKU HRECF-107-013-2). All participants provided written informed consent.

## Experimental design

To measure baseline aerobic fitness, the participants were required to complete a progressive exercise test. This test assessed their peak oxygen consumption (VO$_2$peak). Because cupping was applied to the shoulders and upper back, an arm-crank exercise test was conducted to evaluate the postintervention change. Each participant's initial VO$_2$peak value was arranged in descending order and numbered in sequence. By random decision, players with odd numbers were allocated to the cupping group (*n* = 20), whereas those with even numbers were allocated to the sham group (*n* = 20). This allocation strategy was adopted to ensure equivalent baseline aerobic fitness levels of the 2 groups (Fig 1). To minimize potential bias in subjective assessments, the randomization process was single-blinded, meaning that participants were unaware of their group allocation. However, researchers were informed of the assignments

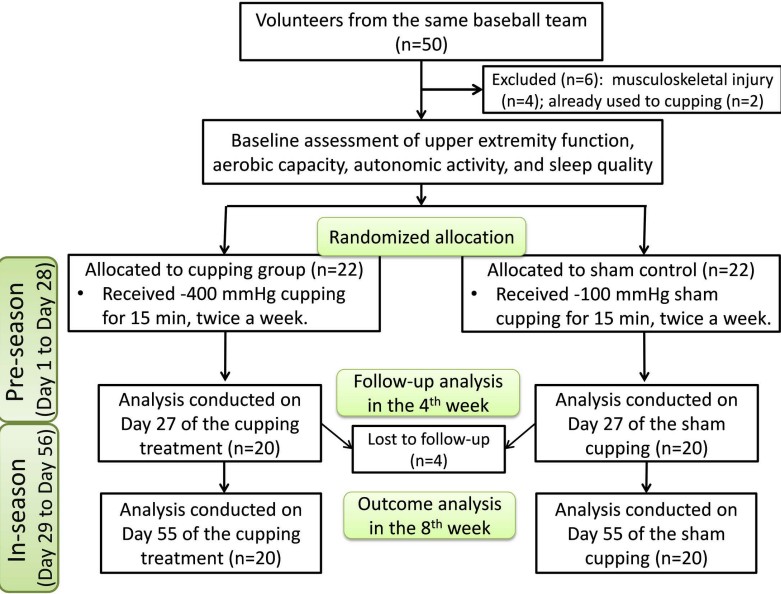

**Fig 1. Flowchart depicting the experimental procedure.**

to ensure the accurate application of the intervention, particularly the cupping pressure. The tests were conducted in an air-conditioned laboratory. The experimental period spanned from 4 weeks before the start of the baseball season (PSC period) to 4 weeks after its commencement (ISC period). The effects of cupping on the athletes' subjective and objective physical performance, autonomic nervous balance, and sleep quality in different periods were investigated.

## Experimental procedure

**Standardized cupping intervention.** The cupping area was the back, and cupping in this area can significantly increase heart rate variability (HRV) activity [15,16]. Baseball players rely on their upper-extremity muscles during throwing and batting. Thus, in addition to five cups being attached to the back, one cup each was attached at the deltoid muscle of the arms (Fig 2). The five cups used on the back had outer and inner diameters of 6.5 and 5.5 cm, respectively. The two cups used on the bilateral deltoid muscle had outer and inner diameters of 5.5 and 4.5 cm, respectively (Shen-Nong Cupping R0416179, Income Instrument, Taiwan). The cupping pressure was set according to that used in previous studies [23]. The cupping group underwent dry cupping at −400 mmHg, whereas the sham group underwent dry cupping at −100 mmHg. The cups in the sham group did not fall off or cause apparent bruising (cupping marks). Cupping at −100 mmHg has been reported to exert no significant effects on ANS activity [15], blood flow near the cup, or exercise performance [14]. The participants underwent 15-min cupping on the upper back and shoulders after routine training or competition. Cupping was performed twice a week, with an interval of 2 to 3 days, for 4 weeks PSC and 4 weeks ISC.

**Functional evaluation of the upper extremities and shoulders.** The Chinese version of the Disabilities of the Arm, Shoulder and Hand (DASH) questionnaire was used to collect data on the participants' upper-extremity function [24], the original scale of which was developed by Sloway et al [25]. This questionnaire comprises a disability/symptom unit (DASHs; 30 items) and a self-selected optional unit that involves highly skilled exercise (DASHe; 4 items). The optional unit is designed for professional athletes and focuses on the unique challenges that they may encounter. To calculate a person's score, all answers were converted into a

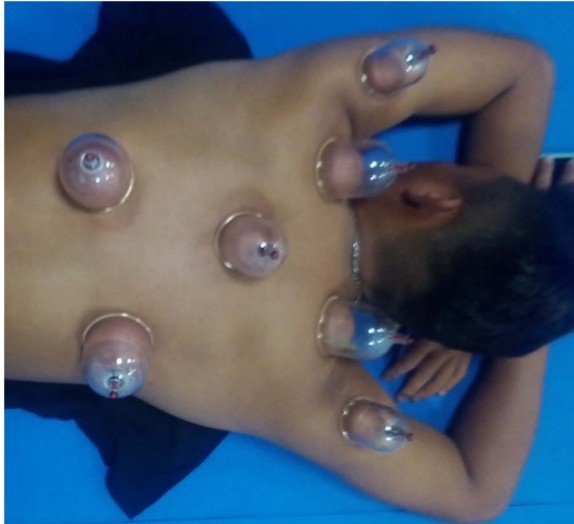

**Fig 2. Placement of 7 cups for dry cupping on a participant's upper back and bilateral deltoid muscle.**

standardized score out of 100 based on the corresponding value. A higher score indicates a higher level of severity.

The Flexilevel Scale of Shoulder Function (FLEX-SF) questionnaire was employed to evaluate the shoulder girdle's function in all aspects of motion [26,27]. This questionnaire comprises a main question, which acquires data for classifying respondents' shoulder joint function into three categories: easy, moderately difficult, and difficult. The participants were required to respond to only items specific to their level of functioning. The maximum total score on this scale is 50 points, and the minimum is 1 point. Higher scores indicate better shoulder girdle function.

**Assessment of the maximal aerobic capacity and ventilatory threshold.** For the arm-crank exercise test, an arm ergometer (ANGIO with a chair, Lode, Groningen, The Netherlands) and a gas analyzer (Vmax 29c; SensorMedics, YorbaLinda, CA, USA) were used. Gas was collected using a mask connected to the gas sampling line and analyzed breath by breath. The test protocol included a warm-up preceding the test; the resistance was increased in a ramp mode of 15 W/min. Throughout the test, the participants were required to maintain the ergometer speed at 60 rpm until they could not maintain it (i.e., the speed dropped to < 50 rpm) or stopped voluntarily. The intervals between the test initiation and the participant's exhaustion, maximum power, and $VO_2$peak were recorded. Their all-out point was the point at which they met two of the following three standard conditions: their $VO_2$ reached a plateau with increasing exercise power, their respiratory exchange ratio exceeded 1.1, or their heart rate reached 85% of the estimated maximum heart rate for someone their age (formula: 220 − age) [28].

This study combined the V-slope and ventilatory equivalence methods to determine the participants' ventilatory threshold (Tvent) values [28]. The V-slope method, which is commonly used to determine Tvent, involves plotting the relationship between $VCO_2$ and $VO_2$. Tvent is determined from the slopes of the curves for these parameters during the increasing movement mode and by identifying an evident offset point. The ventilation equivalence mode is defined as the exercise intensity point at which the oxygen equivalent (minute ventilation [VE]/$VO_2$) sharply increases, although the carbon dioxide equivalent (VE/$VCO_2$) does not, indicating Tvent. If the V-slope method fails, the ventilation equivalent method should be used to determine Tvent.

**HRV evaluation.** Throughout the exercise test, electrocardiography (ECG) data were recorded to analyze the HRV activity of the participants. The HRV activity at the end of the maximal-intensity exercise period and the 10-min recovery period were recorded to measure the rates of HRV activity recovery during PSC and ISC. A miniature physiological signal recorder (TD3; Taiwan Telemedicine Device Company, Kaohsiung, Taiwan) was used to obtain ECG (lead II) data throughout the test. HRV is a measure of the fluctuation in time intervals between successive heartbeats (R–R interval in ECG); it indicates the interplay between sympathetic activity and parasympathetic activity. The TD3 recorder processes HRV data in 32-second (unit) intervals for high frequency and low frequency spectrum analysis. Changes in overall ANS activity (high-intensity exercise period vs. 10-min recovery period) were monitored using units of 5 min. For spectrum analysis, the collected R–R interval ECG signals were subjected to fast Fourier transform (FFT) through frequency domain analysis. FFT is a mathematical algorithm that converts time-domain signals into frequency-domain signals, enabling the analysis of the power spectrum of HRV data. This method is commonly used in HRV analysis to assess ANS activity, as it helps differentiate between sympathetic and parasympathetic influences. The parameters used in the main analysis were the total power (TP) indicating ANS activity, low-frequency power (LF) indicating both sympathetic and parasympathetic modulation, high-frequency power (HF) indicating parasympathetic

activity, LF% indicating sympathetic activity, HF% indicating sympathetic inhibition, and LF/HF ratio indicating sympathovagal balance [29]. The LF component primarily reflects the combined influence of sympathetic and parasympathetic activity, with a predominant association to sympathetic modulation. The LF/HF ratio is commonly used to assess the balance between sympathetic and parasympathetic activity. However, the interpretation of the LF/HF ratio is complex and may not accurately reflect sympathovagal balance due to various influencing factors. In our study, we prioritized LF and LF/HF ratio over other HRV indices based on their widespread application in assessing autonomic function during exercise. We acknowledge the complexities in interpreting these indices and recognize the importance of considering additional HRV parameters, such as HF, to provide a more comprehensive evaluation of autonomic function. Further details on HRV analysis methods can be found in previous reports [14,15,30].

**Assessment of sleep quality.** Sleep quality was assessed using the Chinese version of Pittsburgh Sleep Quality Index (PSQI), which was translated and validated by Tsai et al. in 2005 [31]. The original scale, which was used to measure the sleep quality and disturbances over the past month, was designed by Buysse et al. [32]. This questionnaire comprises 19 items, which assess seven indicators: sleep quality, sleep latency, sleep duration, habitual sleep efficiency, sleep disturbance, sleeping medication use, and daytime dysfunction. Each indicator has a unique scoring standard and is scored from 0 to 3 points. The total score ranges from 0 to 21 points. A total score (for the seven indicators) of > 5 indicates poor sleep quality, whereas a total score of ≤ 5 shows good sleep quality.

## Statistical analysis

Qualitative measures were analyzed using RM-ANOVA to evaluate the pretest and posttest data for both the PSC period (week 1 vs. week 4) and ISC period (week 4 vs. week 8), focusing on the time effect and group × time interactions across various parameters in the cupping and sham groups. The following parameters were analyzed: subjective perception (DASH, FLEX-SF, and PSQI data) and objective physiological response (maximal exercise performance and HRV activity). Performance was measured in terms of peak power, exercise test duration, peak heart rate, $VO_2$peak, and Tvent. In addition to assessing statistical significance, the clinical relevance of the observed changes was considered by evaluating whether the magnitude of improvements aligns with established thresholds reported in the literature. This helped ensure that the findings were interpreted not only in terms of statistical significance but also in their practical implications for athletic performance and recovery. Statistical analyses were performed using PASW Statistics (version 23.0; SPSS Inc., Chicago, IL, USA). Significance was set at $P < .05$, and tendency was analyzed at .05 $< P < .09$.

## Results

### Effects of dry cupping on upper-extremity function

After 4 weeks of cupping during both PSC and ISC, no significant changes were noted in the participants' DASH questionnaire scores for upper-extremity disability and exercise disability or their FLEX-SF questionnaire scores (time effect). Additionally, no significant differences were observed between the groups, nor in the group × time interaction effect during either period (Table 1). Although the group × time interaction effect for upper-extremity disability (DASHs) exhibited significant difference ($P = .045$), no significant changes were noted in the cupping group; however, a decreasing trend in DASHs was observed in the sham group (Table 1). This finding may be attributable to factors unrelated to cupping during the ISC period.

**Table 1. Upper-extremity function after short-term regular cupping (*n* = 40).**

| Upper extremity function | Group | Pre-test | Post-test | P (F) value | | |
|---|---|---|---|---|---|---|
| | | | | Time | Group | Group × time |
| Preseason conditioning (1–4 weeks) | | | | | | |
| DASHs | Sham (n = 20) | 4.19 ± 1.03 | 4.07 ± 0.88 | 0.774 | 0.327 | 0.925 |
| | Cupping (n = 20) | 3.03 ± 1.03 | 2.81 ± 0.88 | (0.084) | (0.985) | (0.009) |
| DASHe | Sham (n = 20) | 4.38 ± 1.99 | 6.56 ± 2.28 | 0.594 | 0.860 | 0.340 |
| | Cupping (n = 20) | 5.31 ± 1.99 | 4.69 ± 2.28 | (0.288) | (0.031) | (0.934) |
| FLEX-SF | Sham (n = 20) | 45.95 ± 1.37 | 45.35 ± 1.22 | 0.604 | 0.969 | 0.951 |
| | Cupping (n = 20) | 45.95 ± 1.41 | 45.47 ± 1.25 | (0.274) | (0.002) | (0.004) |
| In-season conditioning (4–8 weeks) | | | | | | |
| DASHs | Sham (n = 20) | 4.07 ± 0.88 | 2.10 ± 0.72 | 0.066 | 0.816 | 0.045* |
| | Cupping (n = 20) | 2.81 ± 0.88 | 2.89 ± 0.72 | (3.587) | (0.055) | (4.285) |
| DASHe | Sham (n = 20) | 6.56 ± 2.28 | 5.31 ± 2.13 | 0.692 | 0.646 | 0.692 |
| | Cupping (n = 20) | 4.69 ± 2.28 | 4.69 ± 2.13 | (0.160) | (0.215) | (0.160) |
| FLEX-SF | Sham (n = 20) | 45.68 ± 1.24 | 46.32 ± 1.19 | 0.285 | 0.840 | 0.738 |
| | Cupping (n = 20) | 45.7 ± 1.21 | 46.9 ± 1.16 | (1.177) | (0.041) | (0.113) |

Note: DASH: Disabilities of the Arm, Shoulder, and Hand; DASHs: a disability/symptom unit; DASHe: highly skilled exercise; FLEX-SF: Flexilevel Scale of Shoulder Function. Data are presented as mean ± standard error of the mean (SEM). Statistical analysis was performed using RM-ANOVA.

* P < 0.05, indicates significant difference.

## Effects of dry cupping on exercise performance and aerobic capacity

Cupping during PSC did not lead to significant changes in most parameters of the maximal exercise test (time effect); only the peak heart rate decreased significantly (*P* = .041; Table 2). No significant differences were discovered in the between-group and group × time interaction effects for any exercise parameter. Compared with the sham group, the cupping group exhibited increasing trends for VO$_2$max (*P* = .069) and peak ventilation (*P* = .078), although neither difference was significant. Furthermore, no significant difference in the time or group × time interaction effect for Tvent (anaerobic threshold) during PSC was noted between the two groups (Table 2).

Cupping during ISC significantly increased peak ventilation (time effect; *P* = .025; Table 2). Upward trends were noted in exercise duration (*P* = .086), peak power (*P* = .074), and VO$_2$peak (*P* = .068), although the changes were nonsignificant. No significant difference in the group × time interaction effect for any exercise parameter, time effect for Tvent, or group × time interaction effect for Tvent were observed between the groups (Table 2).

## Effects of dry cupping on the rate of recovery of HRV activity after the incremental exercise test

Four weeks of cupping during PSC did not significantly affect the 10-min recovery rate (trial×time interaction effect) of any HRV parameter after high-intensity exercise, and no significant differences were observed between the groups (Table 3). However, considering the between-group difference in HRV recovery (group × trial × time) after cupping for 4 weeks, the rate of LF recovery after intense exercise was significantly higher in the cupping group than in the sham group (*P* = .013; Fig 3; Table 3). A similar, but nonsignificant, trend was noted in the rate of TP recovery (group × trial × time interaction effect; *P* = .061). However, the rate of HF recovery (group × time interaction effect) was significantly lower in the cupping group than in the sham group (*P* = .05; Table 3).

**Table 2. Participants' performance in the arm-crank exercise test after short-term cupping.**

| | Group / Parameters | Sham (n = 20) | | Cupping (n = 20) | | P (F) value | | |
|---|---|---|---|---|---|---|---|---|
| | | Pre-test | Post-test | Pre-test | Post-test | Time | Group | Group × time |
| **Preseason Conditioning** | *Stress test* | | | | | | | |
| | Duration (s) | 423.2 ± 16.6 | 430.1 ± 18.7 | 423.2 ± 16.6 | 445.5 ± 18.7 | 0.141 | 0.741 | 0.433 |
| | Peak power | 102.8 ± 4.1 | 104.3 ± 4.6 | 102.5 ± 4.1 | 108.4 ± 4.6 | 0.123 | 0.738 | 0.343 |
| | VO₂peak (ml/kg/min) | 22.16 ± 1.29 | 20.45 ± 1.14 | 20.57 ± 1.29 | 21.40 ± 1.14 | 0.519 | 0.841 | 0.069# |
| | HRpeak (bpm) | 149.9 ± 4.1 | 142.9 ± 4.6 | 150.7 ± 4.1 | 147.8 ± 4.6 | 0.041* | 0.619 | 0.375 |
| | VEpeak (L/min) | 59.36 ± 3.78 | 58.00 ± 4.29 | 51.80 ± 3.78 | 60.26 ± 4.29 | 0.197 | 0.601 | 0.078# |
| | *Tvent* | | | | | | | |
| | VO₂ (ml/kg/min) | 10.02 ± 0.46 | 9.86 ± 0.50 | 9.33 ± 0.46 | 10.01 ± 0.50 | 0.484 | 0.638 | 0.261 |
| | Workload (Watt) | 52.55 ± 2.25 | 55.00 ± 2.65 | 51.05 ± 2.25 | 56.65 ± 2.65 | 0.081# | 0.978 | 0.487 |
| | HR (bpm) | 109.4 ± 3.2 | 109.4 ± 3.8 | 111.0 ± 3.2 | 108.7 ± 3.8 | 0.659 | 0.917 | 0.659 |
| **In-season Conditioning** | *Stress test* | | | | | | | |
| | Duration (s) | 430.1 ± 18.7 | 454.0 ± 17.6 | 445.5 ± 18.7 | 457.6 ± 17.6 | 0.086# | 0.691 | 0.566 |
| | Peak power | 104.3 ± 4.6 | 110.6 ± 4.3 | 108.4 ± 4.6 | 111.0 ± 4.3 | 0.074# | 0.694 | 0.447 |
| | VO₂peak (ml/kg/min) | 20.45 ± 1.14 | 21.79 ± 1.42 | 21.40 ± 1.14 | 22.78 ± 1.42 | 0.068# | 0.566 | 0.975 |
| | HRpeak (bpm) | 142.9 ± 4.6 | 147.2 ± 4.3 | 147.8 ± 4.6 | 153.3 ± 4.3 | 0.124 | 0.317 | 0.854 |
| | VEpeak (L/min) | 58.00 ± 4.26 | 67.68 ± 4.87 | 61.44 ± 4.37 | 65.45 ± 5.00 | 0.025* | 0.918 | 0.339 |
| | *Tvent* | | | | | | | |
| | VO₂ (ml/kg/min) | 9.86 ± 0.50 | 9.93 ± 0.54 | 10.01 ± 0.50 | 11.03 ± 0.54 | 0.139 | 0.330 | 0.196 |
| | Workload (Watt) | 55.00 ± 2.65 | 51.45 ± 2.61 | 56.65 ± 2.65 | 56.95 ± 2.61 | 0.512 | 0.208 | 0.438 |
| | HR (bpm) | 109.4 ± 3.8 | 104.3 ± 2.9 | 108.7 ± 3.8 | 111.5 ± 2.9 | 0.682 | 0.419 | 0.157 |

Note: VO₂: oxygen consumption; HR: heart rate; VE: minute ventilation; Tvent: ventilatory threshold.

Data are presented as the mean ± SEM. Statistical analysis was performed using RM-ANOVA.

# $0.05 < P < 0.09$, a trend of significant difference;

\* $P < 0.05$, indicates significant difference.

Cupping during ISC significantly improved the rate of LF% recovery after intense exercise (trial×time interaction effect; $P = .05$; Fig 4A; Table 3). Compared with the sham group, the cupping group exhibited a significantly higher LF% recovery rate (group × time interaction effect; $P = .037$; Fig 4B) and LF/HF activity 10 min after intense exercise (group × time interaction effect; $P = .004$; Table 3).

## Effects of dry cupping on sleep quality during PSC and ISC

After 4 weeks of cupping during PSC, no significant difference was observed in the time effect, between-group, or group × time interaction effect for the total PSQI score (Table 4). However, significant improvement was noted in the participants' daytime dysfunction (C7 index) score (time effect; $P = .012$). The degree of improvement was significantly higher in the cupping group than in the sham (group × time interaction effect; $P = .026$). Furthermore, the cupping group exhibited greater improvement trend in sleep latency (C2 index) compared with the sham group ($P = .060$); however, the group × time interaction effect was nonsignificant.

Cupping during ISC significantly improved sleep quality (total PSQI score), sleep latency, and sleep disturbance (time effect; Table 4). However, no significant between-group difference was noted in the group × time interaction effect for any PSQI component.

**Table 3. HRV activity recovery after the maximal exercise test during PSC and ISC.**

| | HRV | week (trial) | Sham (n=20) | | Dry cupping (n=20) | | P value | | | |
|---|---|---|---|---|---|---|---|---|---|---|
| | | | End of exercise | 10min recovery | End of exercise | 10min recovery | Group | Trial × time | Group × time | Group × trial × time |
| Preseason Conditioning (1–4 weeks) | TP | 1st | 4.75±0.18 | 6.66±0.21 | 4.95±0.19 | 6.33±0.22 | 0.663 | 0.141 | 0.091 | 0.061# |
| | [ln(ms²)] | 4th | 4.90±0.17 | 6.76±0.21 | 4.80±0.18 | 6.58±0.22 | | | | |
| | LF | 1st | 3.17±0.24 | 5.94±0.23 | 3.37±0.25 | 5.60±0.24 | 0.728 | 0.292 | 0.285 | 0.013* |
| | [ln(ms²)] | 4th | 3.37±0.24 | 5.97±0.22 | 3.22±0.25 | 5.86±0.23 | | | | |
| | HF | 1st | 1.31±0.24 | 4.88±0.25 | 1.80±0.26 | 4.55±0.26 | 0.795 | 0.506 | 0.050* | 0.149 |
| | [ln(ms²)] | 4th | 1.56±0.22 | 4.99±0.25 | 1.49±0.23 | 4.61±0.26 | | | | |
| | LF/HF | 1st | 1.86±0.13 | 1.07±0.17 | 1.57±0.13 | 1.05±0.18 | 0.854 | 0.994 | 0.108 | 0.841 |
| | (ln ratio) | 4th | 1.81±0.12 | 0.99±0.16 | 1.74±0.13 | 1.25±0.17 | | | | |
| | LF% | 1st | 67.4±3.0 | 69.3±3.0 | 64.5±3.2 | 69.0±3.2 | 0.977 | 0.960 | 0.360 | 0.731 |
| | | 4th | 66.5±3.2 | 67.2±2.9 | 66.1±3.4 | 71.6±3.0 | | | | |
| In-season Conditioning (4–8 weeks) | TP | 4th | 4.90±0.17 | 6.76±0.21 | 4.82±0.17 | 6.57±0.21 | 0.130 | 0.647 | 0.221 | 0.194 |
| | [ln(ms²)] | 8th | 5.00±0.24 | 7.13±0.20 | 4.74±0.24 | 6.36±0.20 | | | | |
| | LF | 4th | 3.37±0.24 | 5.97±0.22 | 3.24±0.24 | 5.87±0.22 | 0.242 | 0.246 | 0.793 | 0.555 |
| | [ln(ms²)] | 8th | 3.40±0.33 | 6.34±0.20 | 3.02±0.33 | 5.76±0.20 | | | | |
| | HF | 4th | 1.56±0.21 | 4.99±0.24 | 1.50±0.21 | 4.56±0.24 | 0.099 | 0.890 | 0.071# | 0.247 |
| | [ln(ms²)] | 8th | 1.52±0.33 | 5.22±0.25 | 1.30±0.33 | 4.15±0.25 | | | | |
| | LF/HF | 4th | 1.81±0.12 | 0.99±0.16 | 1.75±0.12 | 1.31±0.16 | 0.315 | 0.121 | 0.004** | 0.317 |
| | (ln ratio) | 8th | 1.88±0.13 | 1.13±0.14 | 1.72±0.13 | 1.61±0.14 | | | | |
| | LF% | 4th | 66.5±3.3 | 67.2±2.9 | 66.0±3.3 | 72.8±2.9 | 0.360 | 0.050* | 0.037* | 0.481 |
| | | 8th | 65.4±3.3 | 70.0±2.5 | 63.1±3.3 | 78.1±2.5 | | | | |

Note: HRV: heart rate variability; TP: total power; LF: low-frequency power; HF: high-frequency power; LF/HF: ratio of LF to HF; LF%: LF in normalized unit. Data are presented as mean ± SEM. Statistical analysis was performed using RM-ANOVA.

#$0.05 < P < 0.09$, a trend of significant difference;

*$P < 0.05$;

**$P < 0.01$, indicates significant difference.

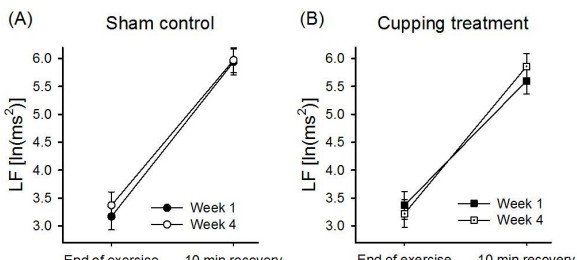

**Fig 3. Recovery of LF activity in the cupping and sham groups after intense exercise.** The rate of LF recovery was significantly higher in the cupping group compared to the sham group after 4 weeks of intervention, as determined by RM-ANOVA, which showed a significant group × trial × time interaction effect (P = .013). (A) Sham group's response did not differ between weeks 4 and 1. (B) Cupping group's postexercise recovery rate was significantly better in week 4 than in week 1.

## Discussion

This study investigating the effects of cupping on athletes during PSC and ISC periods revealed that regular cupping improved the quality of sleep during PSC and the rate of HRV activity recovery after intense exercise. Cupping accelerated the recovery of both sympathetic

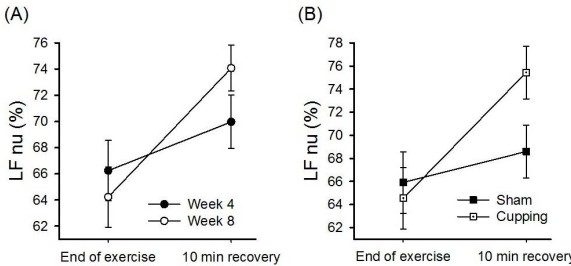

**Fig 4. Effects of cupping on postexercise LF% recovery after strenuous exercise were analyzed using RM-ANOVA.** (A) Cupping for 4 weeks significantly improved the rate of LF% recovery (trial×time interaction effect; $P = .05$) during ISC. (B) Rate of postexercise LF% recovery was significantly better in the cupping group than in the sham group (group × time interaction effect; $P = .037$).

**Table 4. Effects of cupping on sleep quality during PSC and ISC.**

| | Group | Sham (n = 20) | | Dry cupping (n = 20) | | P (F) value | | |
|---|---|---|---|---|---|---|---|---|
| PSQI | | Baseline | 4th week | Baseline | 4th week | Time | Group | Group × time |
| Preseason Conditioning | C1 | 0.95 ± 0.06 | 0.90 ± 0.11 | 0.90 ± 0.06 | 0.65 ± 0.11 | 0.109 | 0.065 | 0.281 |
| | C2 | 1.55 ± 0.30 | 2.10 ± 0.25 | 1.70 ± 0.30 | 1.45 ± 0.25 | 0.472 | 0.450 | 0.060# |
| | C3 | 1.05 ± 0.18 | 1.15 ± 0.15 | 1.20 ± 0.18 | 1.25 ± 0.15 | 0.560 | 0.519 | 0.846 |
| | C4 | 0.00 ± 0.13 | 0.05 ± 0.08 | 0.32 ± 0.13 | 0.11 ± 0.08 | 0.422 | 0.120 | 0.196 |
| | C5 | 5.20 ± 0.81 | 6.00 ± 0.96 | 5.85 ± 0.81 | 6.00 ± 0.96 | 0.435 | 0.770 | 0.593 |
| | C6 | 0 | 0 | 0 | 0 | N/A | N/A | N/A |
| | C7 | 1.35 ± 0.23 | 1.30 ± 0.27 | 1.55 ± 0.23 | 0.80 ± 0.27 | 0.012* | 0.637 | 0.026* |
| | Total | 10.10 ± 1.18 | 11.50 ± 1.28 | 11.50 ± 1.18 | 10.25 ± 1.28 | 0.928 | 0.961 | 0.118 |
| In-season Conditioning | C1 | 0.90 ± 0.11 | 0.75 ± 0.11 | 0.65 ± 0.11 | 0.70 ± 0.11 | 0.531 | 0.268 | 0.214 |
| | C2 | 2.10 ± 0.25 | 1.40 ± 0.27 | 1.45 ± 0.25 | 1.05 ± 0.27 | 0.001** | 0.143 | 0.329 |
| | C3 | 1.15 ± 0.15 | 1.30 ± 0.14 | 1.25 ± 0.15 | 1.25 ± 0.14 | 0.542 | 0.875 | 0.542 |
| | C4 | 0.05 ± 0.08 | 0.10 ± 0.06 | 0.11 ± 0.08 | 0.05 ± 0.06 | 1.000 | 1.000 | 0.492 |
| | C5 | 6.00 ± 0.96 | 4.30 ± 0.85 | 6.00 ± 0.96 | 5.60 ± 0.85 | 0.028* | 0.590 | 0.165 |
| | C6 | 0 | 0 | 0 | 0 | N/A | N/A | N/A |
| | C7 | 1.30 ± 0.27 | 0.85 ± 0.26 | 0.80 ± 0.27 | 0.85 ± 0.26 | 0.320 | 0.436 | 0.216 |
| | Total | 11.5 ± 1.28 | 8.75 ± 1.13 | 10.25 ± 1.28 | 9.50 ± 1.13 | 0.011* | 0.875 | 0.137 |

PSQI, Pittsburgh Sleep Quality Index; C1, sleep quality; C2, sleep latency; C3, sleep duration; C4, habitual sleep efficiency; C5, sleep disturbances; C6, sleep medication; C7, daytime dysfunction; N/A, not available. Data are presented as mean ± SEM. Statistical analysis was performed using RM-ANOVA.

# $0.05 < P < 0.09$, a trend of significant difference;

* $P < 0.05$;

** $P < 0.01$, indicates significant difference.

and parasympathetic modulation during PSC but only that of sympathetic activity during ISC. However, cupping exerted no significant effects on the participants' perceived upper-extremity function, shoulder function, or aerobic capacity.

Cupping therapy is currently used to treat a broad range of medical conditions. Nonetheless, the mechanism of action of cupping remains unclear. A review article compiled many studies in recent years and proposed that several theories have been used to explain the effects of cupping. For example, pain relief and changes in skin biomechanical properties could be explained by the "Pain-Gate Theory," "Diffuse Noxious Inhibitory Controls," and "Reflex Zone Theory." Muscle relaxation, changes in tissue structure, and increases in blood

circulation might be explained by the "Nitric Oxide Theory." The hormonal adjustment and immune effects might be attributed to the "Activation of the Immune System Theory" [33]. However, no single theory explains the whole effects of cupping. In fact, these theories may overlap or interchange with each other to produce different therapeutic effects for certain specific ailments. Owing to the broad and diverse therapeutic effects of cupping, this study applied it to athletes experiencing various physical impacts during PSC and ISC. As expected, we observed that cupping produced different effects on athletes between the PSC and ISC periods.

This study revealed that cupping did not significantly affect the upper-extremity function or shoulder function of the participants during PSC or ISC (Table 1). By contrast, Chen et al. have demonstrated that cupping during the recovery period after strenuous exercise resulted in immediate alleviation of muscle fatigue [14]. Similarly, Hou et al. have reported that cupping significantly reduced fatigue induced by continuous curling of the biceps [13]. A comprehensive review of the literature confirmed that cupping enhances blood flow, increases the oxygenated hemoglobin level in the treated area, facilitates metabolite elimination, and reduces pain, all of which make cupping a promising intervention for musculoskeletal and sports rehabilitation [11]. The discrepancy between the present and previous studies may be attributable to the absence of evident functional impairment in the participants' upper extremities. This inference is supported by the participants' scores on the DASH and FLEX-SF questionnaires. In the present study, the participants' scores on the DASH questionnaire (maximum score: 100 points) ranged from approximately 3 to 6 points. Their scores on the FLEX-SF questionnaire (maximum score: 50 points) ranged from 45 to 46 points. Because of the existence of a ceiling effect, this study was unlikely to discover cupping-mediated significant improvements in upper-extremity function and shoulder function.

This study explored the effects of regular cupping on athletes' performance in a maximal exercise test and observed no significant group-by-time interaction in peak power, $VO_2$peak, or Tvent during PSC or ISC (Table 2). This finding is consistent with that of Chen et al., who investigated the immediate effects of cupping after strenuous exercise, in which no apparent effect was observed on any parameter of the maximal exercise test [14]. In the present study, cupping during PSC tended to increase the cupping group's $VO_2$peak to higher that of the sham group ($P = .069$). This trend may be attributable to the positive effects cupping on blood flow [18] and the oxygenated hemoglobin level [19] in the treated area. The time effect of cupping during ISC generally resulted in increases in peak power and $VO_2$peak ($P = .074$ and.068); however, no such trend was discovered for cupping during PSC. Thus, the participants may have experienced high levels of competition pressure during ISC, likely because of higher morale during ISC than during PSC. However, the aforementioned trend did not achieve statistical significance, and its effect on the participants' actual competition performance might have been limited.

Repeated high-intensity exercise impairs short-term parasympathetic reactivation [34]. HRV parameters are used to analyze the body's stress response during training; these parameters can offer insights into physiological recovery after training [35]. The rate of HRV recovery after exercise is significantly correlated with subsequent exercise performance [9]. Therefore, enhancing the rate of ANS activity recovery after exercise-induced fatigue is crucial for improving exercise performance. Evidence suggests that cupping can restore HRV activity by stimulating the peripheral nervous system to regulate ANS imbalance [15,16].

Cupping during PSC increased the rate of HRV activity recovery after the maximal exercise test ($P < .05$ for LF and TP; Table 3). As shown in Fig 3, after 4 weeks of regular cupping, the LF activity of the cupping group was rapidly regained after high-intensity exercise; however, the sham group exhibited no significant change in the rate of LF recovery after high-intensity

exercise. As shown in Table 3, cupping during ISC primarily affected the LF/HF ratio ($P$ = .004) and LF% ($P$ = .037); this finding indicates that cupping influences the rate of recovery of sympathetic activity. The rate of sympathetic activity recovery after high-intensity exercise was significantly higher in week 8 than in week 4 ($P$ = .050; Fig 4A). Furthermore, the rate of LF% recovery was significantly higher in the cupping group than in the sham group ($P$ = .037; Fig 4B). Therefore, cupping significantly enhanced the recovery of ANS activity after intense exercise, although the effects of cupping differed between the PSC and ISC periods. Cupping improved the rates of recovery of both sympathetic and parasympathetic modulation during PSC but only that of sympathetic activity during ISC. This study postulates that frequent competitions during ISC keep athletes in a state of sympathetic nerve excitement, which results in lower sympathetic activity and a lower recovery rate after high-intensity exercise. Regular cupping can increase the rate of sympathetic activity recovery, as evident from the changes in LF% in the present study, which may contribute to the trend toward improved sports performance during ISC ($P$ = .074 for peak power; Table 2).

The present findings revealed that baseball players typically have poor sleep quality; the participants' total PSQI scores consistently exceeded 5 (average score range: 8.75 to 11.5) during both PSC and ISC. Previous research by Demirel (2016) demonstrated that athletes typically have poorer sleep quality compared to non-athletes [34]. Notably, total PSQI score was consistently higher during PSC than during ISC. This finding is consistent with Mah et al. (2018), who reported that college athletes frequently have poor sleep quality and that these individuals experience poorer sleep quality, which tends to be worse on campus than during travel for competitions [35]. Although review articles have suggested that training, travel, and competition are the primary contributors to sleep disturbances in elite athletes, the main travel-related factor affecting sleep is jet lag [36]. However, the competitions held during the ISC period in this study did not involve jet lag, potentially minimizing its impact on sleep quality. Poor sleep negatively affects cognitive function, physical function, and sports performance [37,38]. Although insomnia is primarily treated using medications, the pharmacological approach often entails side effects [39]. Cupping, a nonpharmacological approach, offers a convenient and safe alternative and is supported by studies demonstrating its efficacy in mitigating insomnia symptoms and enhancing sleep quality [20]. In the present study, cupping significantly improved daytime dysfunction during PSC ($P$ = .026) and exhibited a tendency to improve sleep latency ($P$ = .060). The observed sleep improvements during PSC may be attributed to cupping's ability to promote relaxation and reduce muscle tension, which may contribute to improved overall sleep quality. These effects could be due to mechanisms such as increased blood circulation, muscle relaxation, and modulation of ANS activity, which are particularly beneficial during high-intensity training phases. However, the overall improvement in sleep quality during ISC (time effect; $P$ = .011) might have stemmed from changes in sleep latency ($P$ = .001) and sleep disturbance ($P$ = .028), rather than cupping itself. This observation is likely related to competition stress and exercise-induced fatigue, which can have a stronger influence on sleep quality compared to the effects of cupping. These findings suggest that cupping may be more effective in improving sleep during relatively less stressful PSC period, where relaxation benefits can have a more pronounced effect, while in the ISC period, other factors such as stress and fatigue might overshadow the impact of cupping.

This study has some limitations. First, because athletes from the same team were recruited, the sample size was small. Second, although training modes and intensities are similar among athletes on the same team, these similarities were limited to the PSC period. With the beginning of the ISC period, coaches made tactical arrangements and scheduled tasks for key players. Consequently, during ISC, pressure and fatigue levels were likely to vary across the sample. Thus, when assessing the effects of cupping during ISC, avoiding interference was challenging

because of the disparities in the participants' physiological or psychological status. Third, to avoid disrupting the team's normal schedule and competitions, simultaneous cupping and evaluation sessions could not be scheduled. Typically, evaluations were performed the day after each cupping session. Thus, the residual (next day) rather than the immediate effects of cupping were observed. Fourth, individual variability in response to cupping may have influenced the results, as factors such as physiological differences and recovery capacity can vary among athletes. Fifth, due to the nature of sham cupping, potential placebo effects cannot be entirely ruled out, which may have contributed to subjective perceptions of improvement. Lastly, whether cupping has similar effects when applied to other types of athletes and in different sports seasons remains unclear. Future research is necessary to improve its practical application.

Practical recommendations include integrating cupping as part of a broader recovery strategy tailored to individual athlete needs and monitoring subjective and objective responses over time. Future research should focus on investigating cupping's effects on specific recovery metrics, such as muscle fatigue, heart rate variability, and performance outcomes in athletes from different sports disciplines. Expanding the scope to include long-term interventions and different competitive levels may provide further insights into the practical applications of cupping in sports settings.

## Conclusions

The effects of dry cupping on subjective perceptions and objective physiological responses differed between the PSC and ISC periods. Based on our findings, baseball players should consider adopting cupping to improve sleep quality during PSC, as poor sleep quality is common among elite athletes. Moreover, cupping should be adopted to improve the overall activity of the ANS after high-intensity routine training during PSC and applied after intense competition during ISC to accelerate the recovery of sympathetic activity. Future research could focus on refining cupping protocols, including optimal frequency, duration and cupping pressure, and extending studies to other athlete populations. Additionally, exploring the long-term effects of cupping could provide further insights into its potential benefits. Overall, this study highlights the potential of cupping as a recovery strategy in baseball players.

## Acknowledgments

We thank all participating athletes and baseball team of I-Shou University for administrative assistance.

## Author contributions

**Conceptualization:** Chien-Liang Chen.

**Data curation:** Chien-Liang Chen, Jing-Shia Tang.

**Funding acquisition:** Chien-Liang Chen.

**Investigation:** Jing-Shia Tang.

**Methodology:** Chien-Liang Chen, Jing-Shia Tang.

**Writing – original draft:** Chien-Liang Chen.

**Writing – review & editing:** Chien-Liang Chen, Jing-Shia Tang.

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
