## [Decision Letter · Decision Letter 0]

14 Jan 2025

PONE-D-24-57248Effects of dry cupping on exercise, autonomic activity and sleep in baseball players during preseason and in-season conditioningPLOS ONE

Dear Dr. Chen,

Thank you for submitting your manuscript to PLOS ONE. After careful consideration, we feel that it has merit but does not fully meet PLOS ONE’s publication criteria as it currently stands. Therefore, we invite you to submit a revised version of the manuscript that addresses the points raised during the review process.

We look forward to receiving your revised manuscript.

Kind regards,

Rajib Chowdhury, M.Sc.; MPH

Academic Editor

PLOS ONE

**Journal Requirements:**

The Ministry of Science and Technology of the Republic of China for financially supporting this research under contract MOST 107-2314-B-214-002.

Reviewers' comments:

Reviewer's Responses to Questions

**Comments to the Author**

1. Is the manuscript technically sound, and do the data support the conclusions?

Reviewer #1: Yes

Reviewer #2: Yes

Reviewer #3: Yes

Reviewer #4: Yes

2. Has the statistical analysis been performed appropriately and rigorously? 

Reviewer #1: Yes

Reviewer #2: Yes

Reviewer #3: Yes

Reviewer #4: Yes

3. Have the authors made all data underlying the findings in their manuscript fully available?

Reviewer #1: Yes

Reviewer #2: Yes

Reviewer #3: Yes

Reviewer #4: Yes

4. Is the manuscript presented in an intelligible fashion and written in standard English?

Reviewer #1: Yes

Reviewer #2: Yes

Reviewer #3: Yes

Reviewer #4: Yes

5. Review Comments to the Author

**Reviewer #1: ** The article is exceptionally well-written and thoroughly researched. The author has demonstrated a deep understanding of the subject matter, presenting complex ideas in a clear and concise manner. However, there are several points that need to be addressed:

1. Mention sample size calculations in method section instead of analysis section.

2. Write the name of statistical tests that were applied.

3. Mention how qualitative and quantitative measures were calculated.

4. In tables, mention which tests are applied in legend and in columns, mention if mean/median/IQR/SD were calculated, many readers may not understand without respective headings.

6. Table 3 needs formatting. If possible, present this page in landscape.

7. Line 314 - 318: Cite references for these theories.

**Reviewer #2:**  The manuscript, titled Effects of Dry Cupping on Exercise, Autonomic Activity, and Sleep in Baseball Players During Preseason and In-Season Conditioning, investigates an underexplored area of sports science by assessing the differential impacts of cupping therapy during distinct training phases. The study is thoughtfully designed, methodologically rigorous, and provides insightful findings that contribute to the broader understanding of non-pharmacological recovery methods in sports. Below is an overall assessment of the manuscript’s strengths and areas for improvement:

Abstract

1. The abstract provides a clear summary of the study’s objectives, methods, results, and conclusions. However, it could benefit from slightly tighter phrasing to improve readability. For example: Instead of “Cupping during PSC significantly improved daytime dysfunction scores on the Pittsburgh sleep quality index,” consider “Cupping during PSC notably enhanced daytime function, as measured by the Pittsburgh Sleep Quality Index.”

Introduction

2. I suggest to add a brief mention of other non-pharmacological recovery methods for comparison, enhancing the context for cupping.

3. The research question is clearly articulated. However, the hypothesis could be more explicitly stated.

Methods

4. It would help to specify whether the randomization process was blinded to participants and researchers.

5. A flow diagram could make the experimental design and participant flow more accessible.

6. HRV data collection and analysis are described rigorously. However, briefly explaining why LF and LF/HF were prioritized over other HRV indices would provide clarity.

7. Include post hoc testing methods for significant findings.

8. Highlighting clinically meaningful outcomes, in addition to statistical significance, could add value.

Discussion

9. Consider integrating more mechanistic insights into why sleep improvements were limited to PSC.

10. Strengths are well-highlighted, but limitations could be expanded. For example: Variability in individual responses to cupping; Potential placebo effects due to the nature of sham cupping.

11. Practical recommendations for athletes and coaches are useful. Suggest areas for further research, such as cupping’s effects on recovery metrics in other sports.

Conclusion

12. The conclusion succinctly summarizes findings but could be more forward-looking, emphasizing the potential for refining cupping protocols.

13. Suggest extending studies to other athlete populations or exploring long-term effects.

**Reviewer #3: ** Your article contains all the essential data required for the topic, presented in a comprehensive and well-structured manner. The content is relevant and supports the objectives of the study effectively. I appreciate the clarity and depth of your analysis.

**Reviewer #4: ** Overall, very well written article. Here are a few recommended edits for the manuscript.

On pg 7, line 121, recommend you change 8 weeks to 4 weeks PSC and 4 weeks ISC, or 8 weeks total. The way it is currently written with "for 8 weeks" could be interpreted as 8 weeks consecutively.

On pg 7, lines 136-138, recommend changing the sentence to, "This questionnaire comprises a main question, which acquires data for classifying respondents’ shoulder joint function into three categories: easy, moderately difficult, and difficult."

On pg 8, line 152, recommend changing "exceeded 1.1" to "exceeded 1:1"

On pg 9, line 173, recommend considering explaining what "fast Fourier transform" is and possibly why it is used.

On pg 9, line 184, recommend removing "as follows:" and replace with "seven indicators:"

On pg 10, line 197, is the ISC period weeks 4-8 (or 5 weeks) or 5-8 (4 weeks)? If it is 4-8, then the ISC period is longer than the PSC period and week 4 is an overlap week. Will need to check data to make sure it matches the correct weeks.

On pg 10, line 212; pg 13, line 255; AND pg 15, line 280, recommend "4-week cupping" or "cupping for 4 weeks"

On pg 10, lines 207-217, can these two paragraphs be combined for PSC and ISC when there is no significant difference in data? Then, just focus on where there was a difference.

On page 11, line 221, need a period after the word "period"

On pg 12, line 238, recommend removal of the word "only"

On pg 14, Table 3, the HRV and Week(Trial) columns and the horizontal labeling is difficult to read or follow

On pg 17, line 315, recommend capitalizing "Zone Theory" to match other titles; same with line 318 "activation of the immune system theory"

On pg 17, lines 315-318- are there references for these theories?

On pg 19, lines 381-385, It is written that traveling the sleep quality improves (line 383) and then you state that travel negatively impacts sleep (line 384). This leads to confusion- recommend clarifying this point.

On pg 19, line 402, There is no previous mention in the Methods section that the data collection timeline was different per group, recommend clarifying this in the previous sections.

6. PLOS authors have the option to publish the peer review history of their article (what does this mean? ). If published, this will include your full peer review and any attached files.

**Do you want your identity to be public for this peer review?** For information about this choice, including consent withdrawal, please see our Privacy Policy .

Reviewer #1: **Yes: ** Tabeer Tanwir Awan

Reviewer #2: **Yes: ** Júlio Alejandro Henriques Castro da Costa

Reviewer #3: **Yes: ** Jayantika Bhardwaj

Reviewer #4: No

---

## [Author Response · Author response to Decision Letter 0]

25 Jan 2025

Sites of Revision and Reply to the Comments of the Reviewers

(Manuscript Number: PONE-D-24-57248)

Thank you for your constructive and positive review of our manuscript entitled “Effects of dry cupping on exercise, autonomic activity and sleep in baseball players during preseason and in-season conditioning”. Our responses are included in the following pages. All the changes (included text and references) in the revised manuscript made in response to these comments are clearly indicated with red highlights. We hope that the revisions are in order.

Responses to Reviewers’ Comments

Reviewer : 1

1. Mention sample size calculations in method section instead of analysis section.

Response: We sincerely appreciate the reviewer’s insightful comment. In response to the suggestion, we have revised the manuscript accordingly by relocating the description of the sample size calculations to the first paragraph of the Methods section (Lines 93–99 of the revised version). This adjustment ensures better alignment with the section's overall structure and enhances the clarity of the presentation.

Changes in the manuscript:

The adequacy of the sample size for this study was evaluated using G-Power software (version 3.1.9.4, Heinrich-Heine-Universität Düsseldorf, Germany). An a priori power analysis for repeated measures analysis of variance (RM-ANOVA) was conducted, with the following parameters: an effect size of .60 [22], a statistical power of .80, and an alpha level of .05. The analysis indicated a minimum required sample size of 24 participants. Thus, the final sample of 40 participants exceeded the minimum requirement, supporting the statistical validity of the study.

2. Write the name of statistical tests that were applied.

Response: We sincerely thank the reviewer for their valuable comments. The statistical test applied in this study was repeated measures analysis of variance (RM-ANOVA). This has been described in the estimation of sample size (Line 95) and in the opening line of the Statistical Analysis section (Line 215).

3. Mention how qualitative and quantitative measures were calculated.

Response: Thank you for your thoughtfulness bringing this to our attention. The analysis of questionnaire scales and physiological parameters in this study comprised exclusively quantitative measurements. No qualitative data were collected, as this study did not include qualitative interviews or other qualitative assessment methods. The quantitative analysis methods of this study are described in the statistical analysis section (lines 215-225).

Changes in the manuscript:

Qualitative measures were analyzed using RM-ANOVA to evaluate the pretest and posttest data for both the PSC period (week 1 vs. week 4) and ISC period (week 4 vs. week 8), focusing on the time effect and group × time interactions across various parameters in the cupping and sham groups…………………..This helped ensure that the findings were interpreted not only in terms of statistical significance but also in their practical implications for athletic performance and recovery.

4. In tables, mention which tests are applied in legend and in columns, mention if mean/median/IQR/SD were calculated, many readers may not understand without respective headings.

Response: Thank you for your insightful comment. We have added the statistical tests applied to the data in the captions of Tables 1–4 and Figures 3 and 4. Additionally, we have explicitly stated that the data are presented as mean ± standard error of the mean (SEM). These revisions were implemented to enhance clarity and ensure that readers can readily interpret the data.

Changes in the manuscript:

1). Add the following to the footnotes of Tables 1 to 4: Data are presented as mean ± standard error of the mean (SEM). Statistical analysis was performed using RM-ANOVA.

2). Caption for Fig 3: The rate of LF recovery was significantly higher in the cupping group compared to the sham group after 4 weeks of intervention, as determined by RM-ANOVA, which showed a significant group × trial × time interaction effect (P = .013).

3). Caption for Fig 4: Effects of cupping on postexercise LF% recovery after strenuous exercise were analyzed using RM-ANOVA.

5. Table 3 needs formatting. If possible, present this page in landscape.

Response: Thank you for your suggestion. In response, we have reformatted Table 3 and presented it in landscape orientation as requested. The updated table has been uploaded as supplementary information for better readability.

6. Line 314 - 318: Cite references for these theories.

Response: The theories mentioned in this section are primarily derived from a review article (Reference 33, cited in line 337 and listed in lines 563-565). This article provides a systematic review of the mechanisms of cupping therapy as proposed by recent scholars and summarizes the various mechanisms discussed throughout the text.

Reference 33:

Al-Bedah AMN, et al. The medical perspective of cupping therapy: Effects and mechanisms of action. J Tradit Complement Med. 2018; 9(2): 90-97. doi: 10.1016/j.jtcme.2018.02.003

Reviewer: 2

Abstract

1. The abstract provides a clear summary of the study’s objectives, methods, results, and conclusions. However, it could benefit from slightly tighter phrasing to improve readability. For example: Instead of “Cupping during PSC significantly improved daytime dysfunction scores on the Pittsburgh sleep quality index,” consider “Cupping during PSC notably enhanced daytime function, as measured by the Pittsburgh Sleep Quality Index.”

Response: We are very grateful for the reviewer’s comments. In response, we have revised the sentence in lines 38-39.

Changes in the manuscript:

Cupping during PSC notably enhanced daytime function, as measured by the Pittsburgh Sleep Quality Index.

Introduction

2. I suggest to add a brief mention of other non-pharmacological recovery methods for comparison, enhancing the context for cupping.

Response: Thank you for your valuable suggestion. We agree that providing a brief mention of other non-pharmacological recovery methods would enhance the context for cupping and offer a comparative perspective. In response, we have added a brief discussion of relevant non-pharmacological recovery approaches to lines 60-63 of the revised manuscript. This addition serves to highlight the broader landscape of recovery strategies and situates cupping within this context.

Changes in the manuscript:

It is common for athletes to employ non-pharmacological interventions, such as alternating hot and cold-water therapy, cryotherapy, compression garments, electrical stimulation, and massage, to enhance recovery from training-induced fatigue [Reference 10]. However, these methods may not be as economical or convenient as cupping.

Newly added reference:

[10] Barnett A. Using recovery modalities between training sessions in elite athletes. Sports Med. 2006; 36(9): 781-796. doi.org/10.2165/00007256-200636090-00005

3. The research question is clearly articulated. However, the hypothesis could be more explicitly stated.

Response: Thank you for your valuable feedback. In response, we have explicitly stated our hypothesis in the Introduction (Lines 77-81 of the revised version) to clarify the expected outcomes of the study. We hope this revision addresses your comment effectively.

Changes in the manuscript:

We hypothesized that regular dry cupping applied to the upper body muscles (trapezius and deltoids), commonly used during pitching and batting, would lead to improvements in subjective perceptions, such as sleep quality and upper limb and shoulder function, as well as objective physiological responses related to ANS activity during both the PSC and ISC periods.

Methods

4. It would help to specify whether the randomization process was blinded to participants and researchers.

Response: Thank you for your insightful comment. In response, we clarify that the randomization process was single-blinded. The participants were blinded to their group allocation to minimize potential bias in subjective assessments. However, due to the necessity of applying specific cupping pressures accurately, the researchers were aware of each participant’s assigned group. This approach ensured the correct implementation of the intervention while maintaining blinding on the participants' side to enhance the study's internal validity. We revised the manuscript accordingly to provide this clarification (Lines 109-112).

Changes in the manuscript:

To minimize potential bias in subjective assessments, the randomization process was single-blinded, meaning that participants were unaware of their group allocation. However, researchers were informed of the assignments to ensure the accurate application of the intervention, particularly the cupping pressure.

5. A flow diagram could make the experimental design and participant flow more accessible.

Response: Thank you for your constructive suggestion. In response, we have revised the experimental design and participant flow in the original Figure 1 to enhance clarity and accessibility. A revised flow diagram has been created and included in the updated manuscript to visually represent the study design and participant allocation process, thereby improving the overall readability and comprehensibility of the methodology.

Changes in the manuscript:

6. HRV data collection and analysis are described rigorously. However, briefly explaining why LF and LF/HF were prioritized over other HRV indices would provide clarity.

Response: Thank you for your insightful comment. The prioritization of LF and the LF/HF ratio was based on their established relevance in assessing autonomic nervous system function during exercise, as they are widely used in the literature. We chose these indices for their robust applicability, while also acknowledging the importance of other HRV parameters, such as HF, in providing a comprehensive analysis. Further details on this rationale are provided in lines 194-202 of the manuscript.

Changes in the manuscript:

The LF component primarily reflects the combined influence of sympathetic and parasympathetic activity, with a predominant association to sympathetic modulation. The LF/HF ratio is commonly used to assess the balance between sympathetic and parasympathetic activity. However, the interpretation of the LF/HF ratio is complex and may not accurately reflect sympathovagal balance due to various influencing factors. In our study, we prioritized LF and LF/HF ratio over other HRV indices based on their widespread application in assessing autonomic function during exercise. We acknowledge the complexities in interpreting these indices and recognize the importance of considering additional HRV parameters, such as HF, to provide a more comprehensive evaluation of autonomic function.

7. Include post hoc testing methods for significant findings.

Response: Thank you for your valuable feedback. In our study, we employed repeated measures analysis of variance (RM-ANOVA) to assess the effects of cupping on sleep quality during the PSC and ISC periods. Given that our analysis involved only two groups—cupping and sham—post hoc tests were not conducted. This approach aligns with standard statistical practices, as post hoc analyses are typically reserved for situations involving multiple group comparisons to identify specific group differences.

8. Highlighting clinically meaningful outcomes, in addition to statistical significance, could add value.

Response: Thank you for your valuable suggestion. We acknowledge the importance of considering clinically meaningful outcomes alongside statistical significance to enhance the practical applicability of our findings. In response, we will incorporate a discussion on the clinical relevance of the observed changes in key parameters, such as peak power, exercise duration, and HRV activity. Specifically, we will evaluate whether the magnitude of changes observed in both the cupping and sham groups aligns with established thresholds for clinically meaningful improvements reported in the literature (Lines 221-225). Furthermore, we will contextualize our findings by discussing their potential implications for athletic performance and recovery strategies (See discussion section).

We appreciate your insightful feedback and believe that these additions will provide a more comprehensive interpretation of our results.

Changes in the manuscript:

In addition to assessing statistical significance, the clinical relevance of the observed changes was considered by evaluating whether the magnitude of improvements aligns with established thresholds reported in the literature. This helped ensure that the findings were interpreted not only in terms of statistical significance but also in their practical implications for athletic performance and recovery.

Discussion

9. Consider integrating more mechanistic insights into why sleep improvements were limited to PSC.

Response: Thank you for your valuable suggestion. In response, we have integrated more mechanistic insights into the discussion to explain why sleep improvements were observed primarily during the PSC period (Lines 412-423). We believe this addition provides a clearer understanding of the differential effects of cupping across the two conditioning periods.

Changes in the manuscript:

1). Lines 412-416: The observed sleep improvements during PSC may be attributed to cupping’s ability to promote relaxation and reduce muscle tension, which may contribute to improved overall sleep quality. These effects could be due to mechanisms such as increased blood circulation, muscle relaxation, and modulation of ANS activity, which are particularly beneficial during high-intensity training phases.

2). Lines 418-423: This observation is likely related to competition stress and exercise-induced fatigue, which can have a stronger influence on sleep quality compared to the effects of cupping. These findings suggest that cupping may be more effective in improving sleep during relatively less stressful PSC period, where relaxation benefits can have a more pronounced effect, while in the ISC period, other factors such as stress and fatigue might overshadow the impact of cupping.

10. Strengths are well-highlighted, but limitations could be expanded. For example: Variability in individual responses to cupping; Potential placebo effects due to the nature of sham cupping.

Response: Thank you for your insightful feedback. In response, we have expanded the limitations section to address individual variability in response to cupping and the potential placebo effects associated with the intervention (in Lines 433-437). We believe these additions provide a more comprehensive discussion of the study's constraints.

Changes in the manuscript:

Fourth, individual variability in response to cupping may have influenced the results, as factors such as physiological differences and recovery capacity can vary among athletes. Fifth, due to the nature of sham cupping, potential placebo effects cannot be entirely ruled out, which may have contributed to subjective perceptions of improvement.

11. Practical recommendations for athletes and coaches are useful. Suggest areas for further research, such as cupping’s effects on recovery metrics in other sports.

Response: Thank you for the insightful feedback. In response, we have expanded the limitations section to include practical recommendations for athletes and coaches, as well as suggestions for future research on cupping’s effects across different sports and recovery metrics (Lines 440-446). We believe these additions will enhance the study’s applicability and provide valuable directions for further exploration.

Changes in the manuscript:

Practical recommendations include integrating cupping as part of a broader recovery strategy tailored to individual athlete needs and monitoring subjective and objective responses over time. Future research should focus on investigating cupping’s effects on specific recovery metrics, such as muscle fatigue, heart rate variability, and performance outcomes in athletes from different sports disciplines. Expanding the scope to include long-term interventions and different competit

---

## [Decision Letter · Decision Letter 1]

4 Feb 2025

Effects of dry cupping on exercise, autonomic activity and sleep in baseball players during preseason and in-season conditioning

PONE-D-24-57248R1

Dear Dr. Chen,

We’re pleased to inform you that your manuscript has been judged scientifically suitable for publication and will be formally accepted for publication once it meets all outstanding technical requirements.

Kind regards,

Rajib Chowdhury, M.Sc.; MPH

Academic Editor

PLOS ONE

Additional Editor Comments (optional):

Reviewers' comments:

Reviewer's Responses to Questions

**Comments to the Author**

1. If the authors have adequately addressed your comments raised in a previous round of review and you feel that this manuscript is now acceptable for publication, you may indicate that here to bypass the “Comments to the Author” section, enter your conflict of interest statement in the “Confidential to Editor” section, and submit your "Accept" recommendation.

Reviewer #2: All comments have been addressed

Reviewer #4: All comments have been addressed

2. Is the manuscript technically sound, and do the data support the conclusions?

Reviewer #2: Yes

Reviewer #4: Yes

3. Has the statistical analysis been performed appropriately and rigorously? 

Reviewer #2: Yes

Reviewer #4: Yes

4. Have the authors made all data underlying the findings in their manuscript fully available?

Reviewer #2: Yes

Reviewer #4: Yes

5. Is the manuscript presented in an intelligible fashion and written in standard English?

Reviewer #2: Yes

Reviewer #4: Yes

6. Review Comments to the Author

Reviewer #2: I am happy with the current version of the manuscript.

The authors did a god job to review and answering to all commentaries made by the reviewers.

Reviewer #4: Thank you for completing the edits. The paper is a very interesting and easy read for clinicians. This was a very insightful article. I look forward to seeing it in publication.

7. PLOS authors have the option to publish the peer review history of their article (what does this mean? ). If published, this will include your full peer review and any attached files.

**Do you want your identity to be public for this peer review?** For information about this choice, including consent withdrawal, please see our Privacy Policy .

Reviewer #2: **Yes: ** Júlio Alejandro Henriques Castro da Costa

Reviewer #4: No

---

## [Editor Report · Acceptance letter]

PONE-D-24-57248R1

PLOS ONE

Dear Dr. Chen,

I'm pleased to inform you that your manuscript has been deemed suitable for publication in PLOS ONE. Congratulations! Your manuscript is now being handed over to our production team.

Kind regards,

on behalf of

Dr. Rajib Chowdhury

Academic Editor

PLOS ONE